# Bayesian Model-Agnostic Meta-Learning

**Jaesik Yoon**[*3], **Taesup Kim**[*‡2], **Ousmane Dia**[1], **Sungwoong Kim**[4],
**Yoshua Bengio**[2,5], **Sungjin Ahn**[‡6]

[1]Element AI, [2]MILA Université de Montréal, [3]SAP, [4]Kakao Brain,
[5]CIFAR Senior Fellow, [6]Rutgers University

## Abstract

Due to the inherent model uncertainty, learning to infer Bayesian posterior from a few-shot dataset is an important step towards robust meta-learning. In this paper, we propose a novel Bayesian model-agnostic meta-learning method. The proposed method combines efficient gradient-based meta-learning with nonparametric variational inference in a principled probabilistic framework. Unlike previous methods, during fast adaptation, the method is capable of learning complex uncertainty structure beyond a simple Gaussian approximation, and during meta-update, a novel Bayesian mechanism prevents meta-level overfitting. Remaining a gradient-based method, it is also the first Bayesian model-agnostic meta-learning method applicable to various tasks including reinforcement learning. Experiment results show the accuracy and robustness of the proposed method in sinusoidal regression, image classification, active learning, and reinforcement learning.

## 1 Introduction

Two-year-old children can infer a new category from only one instance (Smith & Slone, 2017). This is presumed to be because during early learning, a human brain develops foundational structures such as the "shape bias" in order to learn the learning procedure (Landau et al., 1988). This ability, also known as *learning to learn* or *meta-learning* (Biggs, 1985; Bengio et al., 1990), has recently obtained much attention in machine learning by formulating it as few-shot learning (Lake et al., 2015; Vinyals et al., 2016). Because, initiating the learning from scratch, a neural network can hardly learn anything meaningful from such a few data points, a learning algorithm should be able to extract the statistical regularity from past tasks to enable warm-start for subsequent tasks.

Learning a new task from a few examples inherently induces a significant amount of uncertainty. This is apparent when we train a complex model such as a neural network using only a few examples. It is also empirically supported by the fact that a challenge in existing few-shot learning algorithms is their tendency to overfit (Mishra et al., 2017). A robust meta-learning algorithm therefore must be able to systematically deal with such uncertainty in order to be applicable to critical problems such as healthcare and self-driving cars. Bayesian inference provides a principled way to address this issue. It brings us not only robustness to overfitting but also numerous benefits such as improved prediction accuracy by Bayesian ensembling (Balan et al., 2015), active learning (Gal et al., 2016), and principled/safe exploration in reinforcement learning (Houthooft et al., 2016). Therefore, developing a Bayesian few-shot learning method is an important step towards robust meta-learning.

Motivated by the above arguments, in this paper we propose a Bayesian meta-learning method, called Bayesian MAML. By introducing Bayesian methods for fast adaptation and meta-update, the proposed method learns to quickly obtain an approximate posterior of a given unseen task and thus

---

[*]Equal contribution, Correspondence to `sungjin.ahn@rutgers.edu`, [‡]Work done also while working at Element AI

provides the benefits of having access to uncertainty. Being an efficient and scalable gradient-based meta-learner which encodes the meta-level statistical regularity in the initial model parameters, our method is the first Bayesian model-agnostic meta-learning method which is thus applicable to various tasks including reinforcement learning. Combining an efficient nonparametric variational inference method with gradient-based meta-learning in a principled probabilistic framework, it can learn complex uncertainty structures while remaining simple to implement.

The main contributions of the paper are as follows. We propose a novel Bayesian method for meta-learning. The proposed method is based on a novel Bayesian fast adaptation method and a new meta-update loss called the Chaser loss. To our knowledge, the Bayesian fast adaptation is the first in meta-learning that provides flexible capability to capture the complex uncertainty curvature of the task-posterior beyond a simple Gaussian approximation. Furthermore, unlike the previous methods, the Chaser loss prevents meta-level overfitting. In experiments, we show that our method is efficient, accurate, robust, and applicable to various problems: sinusoidal regression, image classification, reinforcement learning, and active learning.

## 2 Preliminaries

Consider a model $y = f_\theta(x)$ parameterized by and differentiable w.r.t. $\theta$. Task $\tau$ is specified by a $K$-shot dataset $\mathcal{D}_\tau$ that consists of a small number of training examples, e.g., $K$ pairs $(x_k, y_k)$ per class for classification. We assume that tasks are sampled from a task distribution $\tau \sim p(\mathcal{T})$ such that the sampled tasks share the statistical regularity of the task distribution. A meta-learning algorithm leverages this regularity to improve the learning efficiency of subsequent tasks. The whole *dataset of tasks* is divided into training/validation/test *tasksets*, and the dataset of each task is further divided into task-training/task-validation/task-test *datasets*.

**Model-Agnostic Meta Learning (MAML)** proposed by Finn et al. (2017) is a gradient-based meta-learning framework. Because it works purely by gradient-based optimization without requiring additional parameters or model modification, it is simple and generally applicable to any model as long as the gradient can be estimated.

In Algorithm 1, we briefly review MAML. At each meta-train iteration $t$, it performs: (i) *Task-Sampling*: a mini-batch $\mathcal{T}_t$ of tasks is sampled from the task distribution $p(\mathcal{T})$. Each task $\tau \in \mathcal{T}_t$ provides task-train data $\mathcal{D}_\tau^{\mathrm{trn}}$ and task-validation data $\mathcal{D}_\tau^{\mathrm{val}}$. (ii) *Fast Adaptation* (or *Inner-Update*): the parameter for each task $\tau$ in $\mathcal{T}_t$ is updated by starting from the *current* generic initial model $\theta_0$ and then performing $n$ gradient descent steps on the task-train loss, an operation which we denote by $\mathrm{GD}_n(\theta_0; \mathcal{D}_\tau^{\mathrm{trn}}, \alpha)$ with $\alpha$ being a step size. (iii) *Meta-Update* (or *Outer-Update*): the generic initial parameter $\theta_0$ is updated by gradient descent. The meta-loss is the summation of task-validation losses for all tasks in $\mathcal{T}_t$, i.e., $\sum \mathcal{L}(\theta_\tau; \mathcal{D}_\tau^{\mathrm{val}})$ where the summation is over all $\tau \in \mathcal{T}_t$. At meta-test time, given an unseen test-task $\bar{\tau} \sim p(\mathcal{T})$, starting from the optimized initial model $\theta_0^*$, we obtain a model $\theta_{\bar{\tau}}$ by taking a small number of inner-update steps using $K$-shot *task-training* data $\mathcal{D}_{\bar{\tau}}^{\mathrm{trn}}$. Then, the learned model $\theta_{\bar{\tau}}$ is evaluated on the *task-test* dataset $\mathcal{D}_{\bar{\tau}}^{\mathrm{tst}}$.

**Stein Variational Gradient Descent (SVGD)** (Liu & Wang, 2016) is a recently proposed nonparametric variational inference method. SVGD combines the strengths of MCMC and variational inference. Unlike traditional variational inference, SVGD does not confine the family of approximate distributions within tractable parametric distributions while still remaining a simple algorithm. Also, it converges faster than MCMC because its update rule is deterministic and leverages the gradient of the target distribution. Specifically, to obtain $M$ samples from target distribution $p(\theta)$, SVGD maintains $M$ instances of model parameters, called *particles*. We denote the particles by $\Theta = \{\theta^m\}_{m=1}^M$. At iteration $t$, each particle $\theta_t \in \Theta_t$ is updated by the following rule:

$$\theta_{t+1} \leftarrow \theta_t + \epsilon_t \phi(\theta_t) \quad \text{where} \quad \phi(\theta_t) = \frac{1}{M} \sum_{j=1}^M \left[ k(\theta_t^j, \theta_t) \nabla_{\theta_t^j} \log p(\theta_t^j) + \nabla_{\theta_t^j} k(\theta_t^j, \theta_t) \right], \quad (1)$$

where $\epsilon_t$ is step-size and $k(x, x')$ is a positive-definite kernel. We can see that a particle consults with other particles by asking their gradients, and thereby determines its own update direction. The importance of other particles is weighted according to the kernel distance, relying more on closer particles. The last term $\nabla_{\theta^j} k(\theta^j, \theta^m)$ enforces repulsive force between particles so that they do not collapse to a point. The resulting particles can be used to obtain the posterior predictive distribution $p(y|x, \mathcal{D}^\tau) = \int p(y|x, \theta) p(\theta|\mathcal{D}^\tau) \mathrm{d}\theta \approx \frac{1}{M} \sum_m p(y|x, \theta^m)$ where $\theta^m \sim p(\theta|\mathcal{D}^\tau)$.

| **Algorithm 1** MAML | **Algorithm 2** Bayesian Fast Adaptation |
|---|---|
| Sample a mini-batch of tasks $\mathcal{T}_t$ from $p(\mathcal{T})$ | Sample a mini-batch of tasks $\mathcal{T}_t$ from $p(\mathcal{T})$ |
| **for** each task $\tau \in \mathcal{T}_t$ **do** | **for** each task $\tau \in \mathcal{T}_t$ **do** |
| $\quad \theta_\tau \leftarrow \text{GD}_n(\theta_0; \mathcal{D}_\tau^{\text{trn}}, \alpha)$ | $\quad \Theta_\tau(\Theta_0) \leftarrow \text{SVGD}_n(\Theta_0; \mathcal{D}_\tau^{\text{trn}}, \alpha)$ |
| **end for** | **end for** |
| $\theta_0 \leftarrow \theta_0 - \beta \nabla_{\theta_0} \sum_{\tau \in \mathcal{T}_t} \mathcal{L}(\theta_\tau; \mathcal{D}_\tau^{\text{val}})$ | $\Theta_0 \leftarrow \Theta_0 - \beta \nabla_{\Theta_0} \sum_{\tau \in \mathcal{T}_t} \mathcal{L}_{\text{BFA}}(\Theta_\tau(\Theta_0); \mathcal{D}_\tau^{\text{val}})$ |

A few properties of SVGD are particularly relevant to the proposed method: (i) when the number of particles $M$ equals 1, SVGD becomes standard gradient ascent on the objective $\log p(\theta)$, (ii) under a certain condition, an SVGD-update increasingly reduces the distance between the approximate distribution defined by the particles and the target distribution, in the sense of Kullback-Leibler (KL) divergence (Liu & Wang, 2016), and finally (iii) it is straightforward to apply to reinforcement learning by using Stein Variational Policy Gradient (SVPG) (Liu et al., 2017).

## 3    Proposed Method

### 3.1    Bayesian Fast Adaptation

Our goal is to *learn to infer* by developing an efficient Bayesian gradient-based meta-learning method to efficiently obtain the task-posterior $p(\theta_\tau | \mathcal{D}_\tau^{\text{trn}})$ of a novel task. As our method is in the same class as MAML – in the sense that it encodes the meta-knowledge in the initial model by gradient-based optimization – we first consider the following probabilistic interpretation of MAML with one inner-update step,

$$p(\mathcal{D}_\mathcal{T}^{\text{val}} \mid \theta_0, \mathcal{D}_\mathcal{T}^{\text{trn}}) = \prod_{\tau \in \mathcal{T}} p(\mathcal{D}_\tau^{\text{val}} \mid \theta_\tau' = \theta_0 + \alpha \nabla_{\theta_0} \log p(\mathcal{D}_\tau^{\text{trn}} \mid \theta_0)), \qquad (2)$$

where $p(\mathcal{D}_\tau^{\text{val}}|\theta_\tau') = \prod_{i=1}^{|\mathcal{D}_\tau^{\text{val}}|} p(y_i|x_i, \theta_\tau')$, $\mathcal{D}_\mathcal{T}^{\text{trn}}$ denotes all task-train sets in training taskset $\mathcal{T}$, and $\mathcal{D}_\mathcal{T}^{\text{val}}$ has the same meaning but for task-validation sets. From the above, we can see that the inner-update step of MAML amounts to obtaining task model $\theta_\tau'$ from which the likelihood of the task-validation set $\mathcal{D}_\tau^{\text{val}}$ is computed. The meta-update step is then to perform maximum likelihood estimation of this model w.r.t. the initial parameter $\theta_0$. This probabilistic interpretation can be extended further to applying empirical Bayes to a hierarchical probabilistic model (Grant et al., 2018) as follows:

$$p(\mathcal{D}_\mathcal{T}^{\text{val}} \mid \theta_0, \mathcal{D}_\mathcal{T}^{\text{trn}}) = \prod_{\tau \in \mathcal{T}} \left( \int p(\mathcal{D}_\tau^{\text{val}} \mid \theta_\tau) p(\theta_\tau \mid \mathcal{D}_\tau^{\text{trn}}, \theta_0) \mathrm{d}\theta_\tau \right). \qquad (3)$$

We see that the probabilistic MAML model in Eq. (2) is a special case of Eq. (3) that approximates the task-train posterior $p(\theta_\tau|\theta_0, \mathcal{D}_\tau^{\text{trn}})$ by a point estimate $\theta_\tau'$. That is, $p(\theta_\tau|\mathcal{D}_\tau^{\text{trn}}, \theta_0) = \delta_{\theta_\tau'}(\theta_\tau)$ where $\delta_y(x) = 1$ if $x = y$, and 0 otherwise. To model the task-train posterior which also becomes the prior of task-validation set, Grant et al. (2018) used an isotropic Gaussian distribution with a fixed variance.

"*Can we use a more flexible task-train posterior than a point estimate or a simple Gaussian distribution while maintaining the efficiency of gradient-based meta-learning?*" This is an important question because as discussed in Grant et al. (2018), the task-train posterior of a Bayesian neural network (BNN) trained with a few-shot dataset would have a significant amount of uncertainty which, according to the Bayesian central limit theorem (Le Cam, 1986; Ahn et al., 2012), cannot be well approximated by a Gaussian distribution.

Our first step for designing such an algorithm starts by noticing that SVGD performs deterministic updates and thus gradients can be backpropagated through the particles. This means that we now maintain $M$ initial particles $\Theta_0$ and by obtaining samples from the task-train posterior $p(\theta_\tau|\mathcal{D}_\tau^{\text{trn}}, \Theta_0)$ using SVGD (which is now conditioned on $\Theta_0$ instead of $\theta_0$), we can optimize the following Monte Carlo approximation of Eq. (3) by computing the gradient of the meta-loss $\log p(\mathcal{D}_\mathcal{T}^{\text{val}}|\Theta_0, \mathcal{D}_\mathcal{T}^{\text{trn}})$ w.r.t. $\Theta_0$,

$$p(\mathcal{D}_\mathcal{T}^{\text{val}} \mid \Theta_0, \mathcal{D}_\mathcal{T}^{\text{trn}}) \approx \prod_{\tau \in \mathcal{T}} \left( \frac{1}{M} \sum_{m=1}^{M} p(\mathcal{D}_\tau^{\text{val}} \mid \theta_\tau^m) \right) \quad \text{where} \quad \theta_\tau^m \sim p(\theta_\tau \mid \mathcal{D}_\tau^{\text{trn}}, \Theta_0). \qquad (4)$$

Being updated by gradient descent, it hence remains an efficient meta-learning method while providing a more flexible way to capture the complex uncertainty structure of the task-train posterior than a point estimate or a simple Gaussian approximation.

Algorithm 2 describes an implementation of the above model. Specifically, at iteration $t$, for each task $\tau$ in a sampled mini-batch $\mathcal{T}_t$, the particles initialized to $\Theta_0$ are updated for $n$ steps by applying the SVGD updater, denoted by $\text{SVGD}_n(\Theta_0; \mathcal{D}_\tau^{\text{trn}})$ – the target distribution (the $p(\theta_t^j)$ in Eq. (1) is set to the task-train posterior $p(\theta_\tau|\mathcal{D}_\tau^{\text{trn}}) \propto p(\mathcal{D}_\tau^{\text{trn}}|\theta_\tau)p(\theta_\tau)^1$. This results in task-wise particles $\Theta_\tau$ for each task $\tau \in \mathcal{T}_t$. Then, for the meta-update, we can use the following meta-loss, $\log p(\mathcal{D}_{\mathcal{T}_t}^{\text{val}}|\Theta_0, \mathcal{D}_{\mathcal{T}_t}^{\text{trn}})$

$$\approx \sum_{\tau \in \mathcal{T}_t} \mathcal{L}_{\text{BFA}}(\Theta_\tau(\Theta_0); \mathcal{D}_\tau^{\text{val}}) \ \text{ where } \ \mathcal{L}_{\text{BFA}}(\Theta_\tau(\Theta_0); \mathcal{D}_\tau^{\text{val}}) = \log\left[\frac{1}{M}\sum_{m=1}^{M} p(\mathcal{D}_\tau^{\text{val}}|\theta_\tau^m)\right], \quad (5)$$

Here, we use $\Theta_\tau(\Theta_0)$ to explicitly denote that $\Theta_\tau$ is a function of $\Theta_0$. Note that, by the above model, all the initial particles in $\Theta_0$ are *jointly* updated in such a way as to find the best joint-formation among them. From this optimized initial particles, the task-posterior of a new task can be obtained quickly, i.e., by taking a small number of update steps, and efficiently, i.e, with a small number of samples. We call this Bayesian Fast Adaptation (BFA). The method can be considered a Bayesian ensemble in which, unlike non-Bayesian ensemble methods, the particles interact with each other to find the best formation representing the task-train posterior. Because SVGD with a single particle, i.e., $M = 1$, is equal to gradient ascent, Algorithm 2 reduces to MAML when $M = 1$.

Although the above algorithm brings the power of Bayesian inference to fast adaptation, it can be numerically unstable due to the product of the task-validation likelihood terms. More importantly, for meta-update it is not performing Bayesian inference. Instead, it looks for the initial prior $\Theta_0$ such that SVGD-updates lead to minimizing the empirical loss on task-validation sets. Therefore, like other meta-learning methods, the BFA model can still suffer from overfitting despite the fact that we use a flexible Bayesian inference in the inner update. The reason is somewhat apparent. Because we perform only a small number of inner-updates while the meta-update is based on empirical risk minimization, the initial model $\Theta_0$ can be overfitted to the task-validation sets when we use highly complex models like deep neural networks. Therefore, to become a fully robust meta-learning approach, it is desired for the method to retain the uncertainty during the meta-update as well while remaining an efficient gradient-based method.

### 3.2 Bayesian Meta-Learning with Chaser Loss

Motivated by the above observation, we propose a novel meta-loss. For this, we start by defining the loss as the dissimilarity between approximate task-train posterior $p_\tau^n \equiv p_n(\theta_\tau|\mathcal{D}_\tau^{\text{trn}}; \Theta_0)$ and true task-posterior $p_\tau^\infty \equiv p(\theta_\tau|\mathcal{D}_\tau^{\text{trn}} \cup \mathcal{D}_\tau^{\text{val}})$. Note that $p_\tau^n$ is obtained by taking $n$ fast-adaptation steps from the initial model. Assuming that we can obtain samples $\Theta_\tau^n$ and $\Theta_\tau^\infty$ respectively from these two distributions, the new meta-learning objective can be written as

$$\arg\min_{\Theta_0} \sum_\tau d_p(p_\tau^n \| p_\tau^\infty) \approx \arg\min_{\Theta_0} \sum_\tau d_s(\Theta_\tau^n(\Theta_0) \| \Theta_\tau^\infty). \quad (6)$$

Here, $d_p(p\|q)$ is a dissimilarity between two distributions $p$ and $q$, and $d_s(s_1\|s_2)$ a distance between two sample sets. We then want to minimize this distance using gradient w.r.t. $\Theta_0$. This is to find optimized $\Theta_0$ from which the task-train posterior can be obtained quickly and closely to the true task-posterior. However, this is intractable because we neither have access to the true posterior $p_\tau^\infty$ nor its samples $\Theta_\tau^\infty$.

To this end, we approximate $\Theta_\tau^\infty$ by $\Theta_\tau^{n+s}$. This is done by (i) obtaining $\Theta_\tau^n$ from $p_n(\theta_\tau|\mathcal{D}_\tau^{\text{trn}}; \Theta_0)$ and then (ii) taking $s$ additional SVGD steps with the updated target distribution $p(\theta_\tau|\mathcal{D}_\tau^{\text{trn}} \cup \mathcal{D}_\tau^{\text{val}})$, i.e., augmented with additional observation $\mathcal{D}_\tau^{\text{val}}$. Although it is valid in theory not to augment the leader with the validation set, to help fast convergence we take advantage of it like other meta-learning methods. Note that, because SVGD-updates provide increasingly better approximations of the target

**Algorithm 3** Bayesian Meta-Learning with Chaser Loss (BMAML)

---

1: Initialize $\Theta_0$
2: **for** $t = 0, \dots$ until converge **do**
3:    Sample a mini-batch of tasks $\mathcal{T}_t$ from $p(\mathcal{T})$
4:    **for** each task $\tau \in \mathcal{T}_t$ **do**
5:       Compute chaser $\Theta_\tau^n(\Theta_0) = \text{SVGD}_n(\Theta_0; \mathcal{D}_\tau^{\text{trn}}, \alpha)$
6:       Compute leader $\Theta_\tau^{n+s}(\Theta_0) = \text{SVGD}_s(\Theta_\tau^n(\Theta_0); \mathcal{D}_\tau^{\text{trn}} \cup \mathcal{D}_\tau^{\text{val}}, \alpha)$
7:    **end for**
8:    $\Theta_0 \leftarrow \Theta_0 - \beta \nabla_{\Theta_0} \sum_{\tau \in \mathcal{T}_t} d_s(\Theta_\tau^n(\Theta_0) \parallel \text{stopgrad}(\Theta_\tau^{n+s}(\Theta_0)))$
9: **end for**

---

distribution as $s$ increases, the *leader* $\Theta_\tau^{n+s}$ becomes closer to the target distribution $\Theta_\tau^\infty$ than the *chaser* $\Theta_\tau^n$. This gives us the following meta-loss:

$$\mathcal{L}_{\text{BMAML}}(\Theta_0) = \sum_{\tau \in \mathcal{T}_t} d_s(\Theta_\tau^n \parallel \Theta_\tau^{n+s}) = \sum_{\tau \in \mathcal{T}_t} \sum_{m=1}^{M} \|\theta_\tau^{n,m} - \theta_\tau^{n+s,m}\|_2^2. \tag{7}$$

Here, to compute the distance between the two sample sets, we make a one-to-one mapping between the leader particles and the chaser particles and compute the Euclidean distance between the paired particles. Note that we do not back-propagate through the leader particles because we use them as targets that the chaser particles follow. A more sophisticated method like maximum mean discrepancy (Borgwardt et al., 2006) can also be used here. In our experiments, setting $n$ and $s$ to a small number like $n = s = 1$ worked well.

Minimizing the above loss w.r.t. $\Theta_0$ places $\Theta_0$ in a region where the chaser $\Theta_\tau^n$ can efficiently *chase* the leader $\Theta_\tau^{n+s}$ in $n$ SVGD-update steps starting from $\Theta_0$. Thus, we call this meta-loss the *Chaser* loss. Because the leader converges to the posterior distribution instead of doing empirical risk minimization, it retains a proper level of uncertainty and thus prevents from meta-level overfitting. In Algorithm 3, we describe the algorithm for supervised learning. One limitation of the method is that, like other ensemble methods, it needs to maintain $M$ model instances. Because this could sometimes be an issue when training a large model, in the Experiment section we introduce a way to share parameters among the particles.

## 4 Related Works

There have been many studies in the past that formulate meta-learning and learning-to-learn from a probabilistic modeling perspective (Tenenbaum, 1999; Fe-Fei et al., 2003; Lawrence & Platt, 2004; Daumé III, 2009). Since then, the remarkable advances in deep neural networks (Krizhevsky et al., 2012; Goodfellow et al., 2016) and the introduction of new few-shot learning datasets (Lake et al., 2015; Ravi & Larochelle, 2017), have rekindled the interest in this problem from the perspective of deep networks for few-shot learning (Santoro et al., 2016; Vinyals et al., 2016; Snell et al., 2017; Duan et al., 2016; Finn et al., 2017; Mishra et al., 2017). Among these, Finn et al. (2017) proposed MAML that formulates meta-learning as gradient-based optimization.

Grant et al. (2018) reinterpreted MAML as a hierarchical Bayesian model, and proposed a way to perform an implicit posterior inference. However, unlike our proposed model, the posterior on validation set is approximated by local Laplace approximation and used a relatively complex 2nd-order optimization using K-FAC (Martens & Grosse, 2015). The fast adaptation is also approximated by a simple isotropic Gaussian with fixed variance. As pointed by Grant et al. (2018), this approximation would not work well for skewed distributions, which is likely to be the case of BNNs trained on a few-shot dataset. The authors also pointed that their method is limited in that the predictive distribution over new data-points is approximated by a point estimate. Our method resolves these limitations. Although it can be expensive when training many large networks, we mitigate this cost by parameter sharing among the particles. In addition, Bauer et al. (2017) also proposed Gaussian approximation of the task-posterior and a scheme of splitting the feature network and the classifier which is similar to what we used for the image classification task. Lacoste et al. (2017) proposed learning a distribution of stochastic input noise while fixing the BNN model parameter.

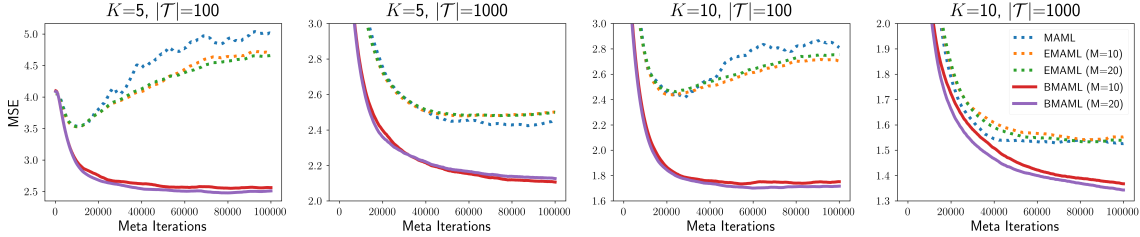

**Figure 1:** Sinusoidal regression experimental results (meta-testing performance) by varying the number of examples ($K$-shot) given for each task and the number of tasks $|\mathcal{T}|$ used for meta-training.

## 5 Experiments

We evaluated our proposed model (BMAML) in various few-shot learning tasks: sinusoidal regression, image classification, active learning, and reinforcement learning. Because our method is a Bayesian ensemble, as a baseline model we used an ensemble of independent MAML models (EMAML) from which we can easily recover regular MAML by setting the number of particles to 1. In all our experiments, we configured BMAML and EMAML to have the same network architecture and used the RBF kernel. The experiments are designed in such a way to see the effects of uncertainty in various ways such as accuracy, robustness, and efficient exploration.

**Regression:** The population of the tasks is defined by a sinusoidal function $y = A\sin(wx + b) + \epsilon$ which is parameterized by amplitude $A$, frequency $w$, and phase $b$, and observation noise $\epsilon$. To sample a task, we sample the parameters uniformly randomly $A \in [0.1, 5.0]$, $b \in [0.0, 2\pi]$, $w \in [0.5, 2.0]$ and add observation noise from $\epsilon \sim \mathcal{N}(0, (0.01A)^2)$. The $K$-shot dataset is obtained by sampling $x$ from $[-5.0, 5.0]$ and then by computing its corresponding $y$ with noise $\epsilon$. Note that, because of the highly varying frequency and observation noise, this is a more challenging setting containing more uncertainty than the setting used in Finn et al. (2017). For the regression model, we used a neural network with 3 layers each of which consists of 40 hidden units.

In Fig. 1, we show the mean squared error (MSE) performance on the test tasks. To see the effect of the degree of uncertainty, we controlled the number of training tasks $|\mathcal{T}|$ to 100 and 1000, and the number of observation shots $K$ to 5 and 10. The lower number of training tasks and observation shots is expected to induce a larger degree of uncertainty. We observe, as we claimed, that both MAML (which is EMAML with $M = 1$) and EMAML overfit severely in the settings with high uncertainty although EMAML with multiple particles seems to be slightly better than MAML. BMAML with the same number of particles provides significantly better robustness and accuracy for all settings. Also, having more particles tends to improve further.

**Classification:** To evaluate the proposed method on a more complex model, we test the performance on the *mini*Imagenet classification task (Vinyals et al., 2016) involving task adaptation of 5-way classification with 1 shot. The dataset consists of 60,000 color images of 84×84 dimension. The images consist of total 100 classes and each of the classes contains 600 examples. The entire classes are split into 64, 12, and 24 classes for meta-train, meta-validation, and meta-test, respectively. We generated the tasks following the same procedure as in Finn et al. (2017).

In order to reduce the space and time complexity of the ensemble models (i.e., BMAML and EMAML) in this large network setting, we used the following parameter sharing scheme among the particles, similarly to Bauer et al. (2017). We split the network architecture into the feature extractor layers and the classifier. The feature extractor is a convolutional network with 5 hidden layers with 64 filters. The classifier is a single-layer fully-connected network with softmax output. The output of the feature extractor which has 256 dimensions is input to the classifier. We share the feature extractor across all the particles while each particle has its own classifier. Therefore, the space complexity of the network is $\mathcal{O}(|\theta_{\text{feature}}| + M|\theta_{\text{classifier}}|)$. Both the classifier and feature extractor are updated during meta-update, but for inner-update only the classifier is updated. The baseline models are updated in the same manner. We describe more details of the setting in Appendix A.2.

We can see from Fig. 2 (a) that for both $M = 5$ and $M = 10$ BMAML provides more accurate predictions than EMAML. However, the performance of both BMAML and EMAML with 10

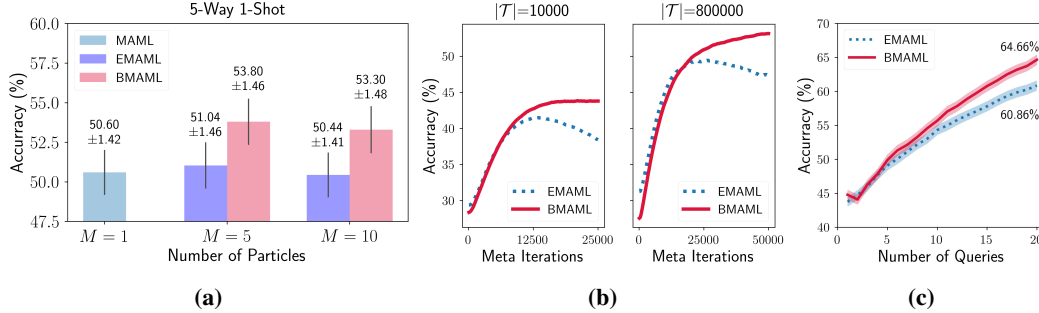

**Figure 2:** Experimental results in *mini*Imagenet dataset: (a) few-shot image classification using different number of particles, (b) using different number of tasks for meta-training, and (c) active learning setting.

particles is slightly lower than having 5 particles[2]. Because a similar instability is also observed in the SVGD paper (Liu & Wang, 2016), we presume that one possible reason is the instability of SVGD such as sensitivity to kernel function parameters. To increase the inherent uncertainty further, in Fig. 2 (b), we reduced the number of training tasks $|\mathcal{T}|$ from 800K to 10K. We see that BMAML provides robust predictions even for such a small number of training tasks while EMAML overfits easily.

**Active Learning:** In addition to the ensembled prediction accuracy, we can also evaluate the effectiveness of the measured uncertainty by applying it to active learning. To demonstrate, we use the *mini*Imagenet classification task. To do this, given an unseen task $\tau$ at test time, we first run a fast adaptation from the meta-trained initial particles $\Theta_0^*$ to obtain $\Theta_\tau$ of the task-train posterior $p(\theta_\tau | \mathcal{D}_\tau; \Theta_0^*)$. For this we used 5-way 1-shot labeled dataset. Then, from a pool of unlabeled data $\mathcal{X}_\tau = \{x_1, \ldots, x_{20}\}$, we choose an item $x^*$ that has the maximum predictive entropy $\arg\max_{x \in \mathcal{X}_\tau} \mathbb{H}[y|x, D_\tau] = -\sum_{y'} p(y'|x, D_\tau) \log p(y'|x, D_\tau)$. The chosen item $x^*$ is then removed from $\mathcal{X}_\tau$ and added to $\mathcal{D}_\tau$ along with its label. We repeat this process until we consume all the data in $\mathcal{X}_\tau$. We set $M$ to 5. As we can see from Fig. 2 (c), active learning using the Bayesian fast adaptation provides consistently better results than EMAML. Particularly, the performance gap increases as more examples are added. This shows that the examples picked by BMAML so as to reduce the uncertainty, provides proper discriminative information by capturing a reasonable approximation of the task-posterior. We presume that the performance degradation observed in the early stage might be due to the class imbalance induced by choosing examples without considering the class balance.

**Reinforcement Learning:** SVPG is a simple way to apply SVGD to policy optimization. Liu et al. (2017) showed that the maximum entropy policy optimization can be recast to Bayesian inference. In this framework, the particle update rule (a particle is now parameters of a policy) is simply to replace the target distribution $\log p(\theta)$ in Eq. (1) with the objective of the maximum entropy policy optimization, i.e., $\mathbb{E}_{q(\theta)}[J(\theta)] + \eta \mathbb{H}[q]]$ where $q(\theta)$ is a distribution of policies, $J(\theta)$ is the expected return of policy $\theta$, and $\eta$ is a parameter for exploration control. Deploying multiple agents (particles) with a principled Bayesian exploration mechanism, SVPG encourages generating diverse policy behaviours while being easy to parallelize.

We test and compare the models on the same MuJoCo continuous control tasks (Todorov et al., 2012) as are used in Finn et al. (2017). In the goal velocity task, the agent receives higher rewards as its current velocity approaches the goal velocity of the task. In the goal direction task, the reward is the magnitude of the velocity in either the forward or backward direction. We tested these tasks for two simulated robots, the ant and the cheetah. The goal velocity is sampled uniformly at random from $[0.0, 2.0]$ for the cheetah and from $[0.0, 3.0]$ for the ant. As the goal velocity and the goal direction change per task, a meta learner is required to learn a given unseen task after trying $K$ episodes. We implemented the policy network with two hidden-layers each with 100 ReLU units. We tested the number of particles for $M \in \{1, 5, 10\}$ with $M = 1$ only for non-ensembled MAML. We describe more details of the experiment setting in Appendix C.1.

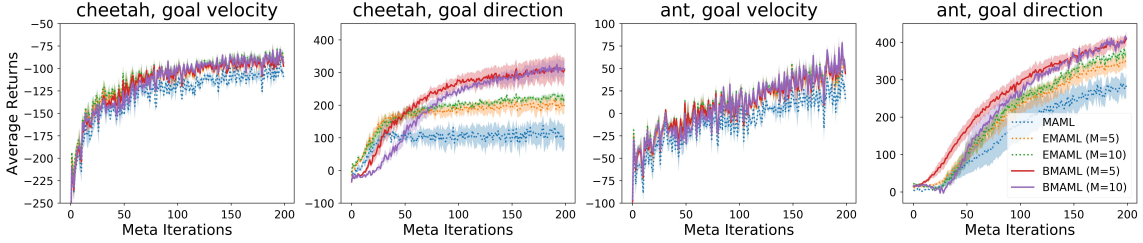

**Figure 3:** Locomotion comparison results of SVPG-TRPO and VPG-TRPO

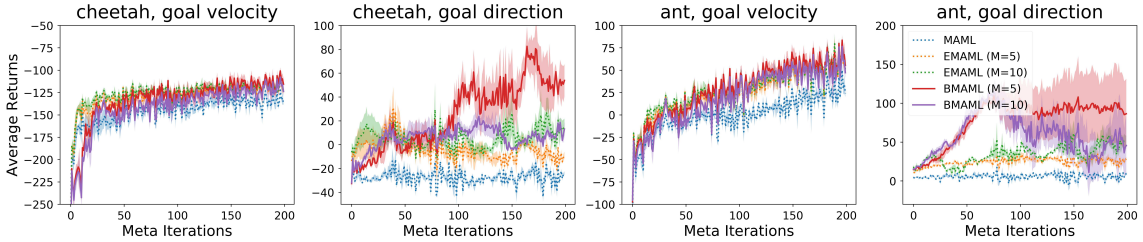

**Figure 4:** Locomotion comparison results of SVPG-Chaser and VPG-Reptile

For meta-update, MAML uses TRPO (Schulman et al., 2015) which is designed with a special purpose to apply for reinforcement learning and uses a rather expensive $2^{nd}$-order optimization. However, the meta-update by the chaser loss is general-purpose and based on $1^{st}$-order optimization[3]. Thus, for a fair comparison, we consider the following two experiment designs. First, in order to evaluate the performance of the inner updates using Bayesian fast adaptation, we compare the standard MAML, which uses vanilla policy gradient (REINFORCE, Williams (1992)) for inner-updates and TRPO for meta-updates, with the Bayesian fast adaptation with TRPO meta-update. We label the former as VPG-TRPO and the later as SVPG-TRPO. Second, we compare SVPG-Chaser with VPG-Reptile. Because, similarly to the chaser loss, Reptile (Nichol et al., 2018) performs $1^{st}$-order gradient optimization based on the distance in the model parameter space, this provides us a fair baseline to evaluate the chaser loss in RL. The VPG-TRPO and VPG-Reptile are implemented with independent multiple agents. We tested the comparing methods for $M = [1, 5, 10]$. More details of the experimental setting is provided in Appendix C.3.

As shown in Fig. 3 and Fig. 4, we can see that overall, BMAML shows superior performance to EMAML. In particular, BMAML performs significantly and consistently better than EMAML in the case of using TRPO meta-updater. In addition, we can see that BMAML performs much better than EMAML for the goal direction tasks. We presume that this is because in the goal direction task, there is no goal velocity and thus a higher reward can always be obtained by searching for a better policy. This, therefore, demonstrates that BMAML can learn a better exploration policy than EMAML. In contrast, in the goal velocity task, exploration becomes less effective because it is not desired once a policy reaches the given goal velocity. This thus explains the results on the goal velocity task in which BMAML provides slightly better performance than EMAML. For some experiments, we also see that having more particles do not necessarily provides further improvements. As in the case of classification, we hypothesize that one of the reasons could be due to the instability of SVGD. In Appendix C.4, we also provide the results on 2D Navigation task, where we observe similar superiority of BMAML to EMAML.

## 6 Discussions

In this section, we discuss some of the issues underlying the design of the proposed method.

*BMAML is tied to SVGD?* In principle, it could actually be more generally applicable to any inference algorithm that can provide *differentiable samples*. Gradient-based MCMC methods like HMC (Neal et al., 2011) or SGLD (Welling & Teh, 2011) are such methods. We however chose SVGD specifically

for BMAML because jointly updating the particles altogether is more efficient for capturing the distribution quickly by a small number of update steps. In contrast, MCMC would require to wait for much more iterations until the chain mixes enough and a long backpropagation steps through the chain.

*Parameter space v.s. prediction space?*  We defined the chaser loss by the distance in the model parameter space although it is also possible to define it in the prediction distance, i.e., by prediction error. We chose the parameter space because (1) we can save computation for the forward-pass for predictions, and (2) it empirically showed better performance for RL and similar performance for other tasks. The advantages of working in the parameter space is also discussed in Nichol et al. (2018).

*Do the small number of SVGD steps converge to the posterior?* In our small-data-big-network setting, a large area of a true task-posterior will be meaningless for other tasks. Thus, it is not desired to fully capture the task-posterior but instead we need to find an area which will be broadly useful for many tasks. This is the goal of hierarchical Bayes which our method approximate by finding such area and putting $\Theta_0$ there. In theory, the task-posterior can be fully captured with infinite number of particles and update-steps, and thus dilute the initialization effect. In practice, the full coverage would, however, not be achievable (and not desired) because SVGD or MCMC would have difficulties in covering all areas of the complex multimodal task-posterior like that of a neural network.

## 7   Conclusion

Motivated by the hierarchical probabilistic modeling perspective to gradient-based meta-learning, we proposed a Bayesian gradient-based meta learning method. To do this, we combined the Stein Variational Gradient Descent with gradient-based meta learning in a probabilistic framework, and proposed the Bayesian Fast Adaptation and the Chaser loss for meta-update. As it remains a model-agnostic model, in experiments, we evaluated the method in various types of learning tasks including supervised learning, active learning, and reinforcement learning, and showed its superior performance in prediction accuracy, robustness to overfitting, and efficient exploration.

As a Bayesian ensemble method, along with its advantages, the proposed method also inherits the generic shortcomings of ensemble methods, particularly the space/time complexity proportional to the number of particles. Although we showed that our parameter sharing scheme is effective to mitigate this issue, it would still be interesting to improve the efficiency further in this direction. In addition, because the performance of SVGD can be sensitive to the parameters of the kernel function, incorporating the fast-adaptation of the kernel parameter into a part of meta-learning would also be an interesting future direction.

### Acknowledgments

JY thanks SAP and Kakao Brain for their support. TK thanks NSERC, MILA and Kakao Brain for their support. YB thanks CIFAR, NSERC, IBM, Google, Facebook and Microsoft for their support. SA, Element AI Fellow, thanks Nicolas Chapados, Pedro Oliveira Pinheiro, Alexandre Lacoste, Negar Rostamzadeh for helpful discussions and feedback.

## Footnotes

[1]In our experiments, we put hyperprior on the variance of the prior (mean is set to 0). Thus, the posterior of hyperparameter is automatically learned also by SVGD, i.e., the particle vectors include the prior parameters.

[2]We found a similar instability in the relationship between the number of particles and the prediction accuracy from the original implementation by the authors of the SVGD paper.

[3]When considering the inner update together, TRPO, Chaser and Reptile are $3^{rd}/2^{nd}/1^{st}$-order, respectively.

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
