[Supplementary Material · BML_NIPS_18 (1)_supplement.pdf]

# Appendix A Supervised Learning

## A.1 Regression

For regression, we used 10 tasks for each meta-batch and the meta-validation dataset $\mathcal{D}_\tau^{\text{val}}$ is set to have the same size of the meta-training dataset $\mathcal{D}_\tau^{\text{trn}}$ ($|\mathcal{D}_\tau^{\text{trn}}| = |\mathcal{D}_\tau^{\text{val}}| = K$). During training, the number of steps $n$ for chaser is set to $n = 1$ and also the number of steps $s$ for leader is set to $s = 1$. We used different step sizes $\alpha$ for computing chaser and leader, 0.01 and 0.001, respectively. This allows the leader to stay nearby the chaser but toward the target posterior and stabilized the training. The models were trained with using different size of training dataset $|\mathcal{T}|$, the number of tasks observable during training, and we trained the model over 10000 epochs for $|\mathcal{T}| = 100$ and 1000 epochs for $|\mathcal{T}| = 1000$. In Fig. 6, we show the qualitative results on randomly sampled sinusoid task and we used 5 update steps. The task-train posterior $p(\theta_\tau | \mathcal{D}_\tau^{\text{trn}})$ decomposes into the train data likelihood and parameter prior as $p(\theta_\tau | \mathcal{D}_\tau^{\text{trn}}) \propto p(\mathcal{D}_\tau^{\text{trn}} | \theta_\tau) p(\theta_\tau)$ and this is formulated as:

$$p(\theta_\tau | \mathcal{D}_\tau^{\text{trn}}) \propto \prod_{(x,y) \in \mathcal{D}_\tau^{\text{trn}}} \mathcal{N}(y | f_W(x), \gamma^{-1}) \prod_{w \in W} \mathcal{N}(w | 0, \lambda^{-1}) \text{Gamma}(\gamma | a, b) \text{Gamma}(\lambda | a', b')$$

where $\theta_\tau$ consists of network parameters $W$ and scaling parameters $\gamma, \lambda$. In all experiments, we set Gamma distribution hyper-parameters as $a = 2.0, b = 0.2$ and $a' = 2.0, b' = 2.0$. During meta-update with chaser-loss, we used Adam optimizer (Kingma & Ba, 2014) with learning rate $\beta = 0.001$.

## A.2 Classification

All models and experiments on the *mini*Imagenet classification task are trained with using the same network architecture and 16 tasks are used for each meta-batch during training. Each task is defined by randomly selected 5 classes with one instance of each class to adapt the model and it is evaluated on unseen instances within the selected 5 classes. We used the meta-validation dataset $\mathcal{D}_\tau^{\text{val}}$ containing one example per each class for the 5-way 1-shot setting. This reduced the computational cost and also improved the performance of all models. During training, the number of steps for chaser and leader both are set to 1 ($n = s = 1$). The chaser and leader used step size $\alpha = 0.01$ and $\alpha = 0.005$, respectively. The meta-update was done by using Adam optimizer ($\beta = 0.0005$). The models were trained with using different size of training dataset $|\mathcal{T}|$ and we trained the model with $|\mathcal{T}| = 800000$ and 1 epoch. With $|\mathcal{T}| = 10000$, the model was trained over 40 epochs. The task-train posterior $p(\theta_\tau | \mathcal{D}_\tau^{\text{trn}})$ for classification is slightly different to the regression task due to using softmax for the data likelihood.

$$p(\theta_\tau | \mathcal{D}_\tau^{\text{trn}}) \propto \prod_{(x,y) \in \mathcal{D}_\tau^{\text{trn}}} p(y | f_W(x)) \prod_{w \in W} \mathcal{N}(w | 0, \lambda^{-1}) \text{Gamma}(\lambda | a, b)$$

where the hyper-parameters for Gamma distribution were set as $a = 2.0, b = 0.2$ or $a = 1.0, b = 0.1$ in our experiments.

# Appendix B Active Learning

---

**Algorithm 4** Active Learning on Image Classification

---

1: Sample a few-shot labeled dataset $\mathcal{D}_\tau$ and a pool of unlabeled dataset $\mathcal{X}_\tau$ of task $\tau$
2: Initialize $\Theta_\tau \leftarrow \Theta_0^*$
3: Update $\Theta_\tau \leftarrow \text{SVGD}_n(\Theta_\tau; \mathcal{D}_\tau, \alpha)$
4: **while** $\mathcal{X}_\tau$ is not empty **do**
5:    Select $x' \leftarrow \arg\max_{x \in \mathcal{X}_\tau} \mathbb{H}[y | x, \Theta_\tau]$ and remove $x'$ from $\mathcal{X}_\tau$
6:    Request $y'$ of $x'$
7:    Update $\mathcal{D}_\tau \leftarrow \mathcal{D}_\tau \cup \{(x', y')\}$
8:    Update $\Theta_\tau \leftarrow \text{SVGD}_n(\Theta_\tau; \mathcal{D}_\tau, \alpha)$
9: **end while**

---

# Appendix C   Reinforcement Learning

## C.1   Locomotion

The locomotion experiments require two simulated robots, a planar cheetah and 3D quadruped ones (called as ant), and two individual goals, to run in a particular direction or at a particular velocity. For the ant goal velocity, a positive bonus reward at each timestep is added to prevent the ant from ending the episode. In those experiments, the timestep in each episodes is 200, the number of episode per each inner update, $K$ is 10 except the ant goal direction task, in which 40 episodes for each inner update is used, because of task complexity. The number of tasks per each meta update is 20, and the models are trained for up to 200 meta iterations.

## C.2   Used Methods

We evaluate our proposed method on two cases, SVPG-TRPO vs VPG-TRPO and SVPG-Chaser vs VPG-Reptile. We describe the methods in this subsection, except VPG-TRPO, because this is MAML when $M = 1$.

### C.2.1   SVPG-TRPO

This method is to use SVPG as inner update and TRPO as meta update, which is following to a simple Bayesian meta-learning manner. In $K$-shot reinforcement learning on this method, $K$ episodes from each policy particles and task $\tau$ (total number of episode is $KM$), and the corresponding rewards are used for task learning on the task. This method gets the above data ($\mathcal{D}_\tau^{trn}$) from $\Theta_0$, and updates the parameters $\Theta_0$ to $\Theta_\tau^n$ with $\mathcal{D}_\tau^{trn}$ and SVPG. After getting few-shot learned parameters ($\Theta_\tau^n$), our method get new data ($\mathcal{D}_\tau^{val}$) from $\Theta_\tau^n$. After all the materials for meta learning have been collected, our method finds the meta loss with few-shot learned particles and task-validation set, $\mathcal{D}_\tau^{val}$. On meta-learning, TRPO (Schulman et al., 2015) is used as MAML (Finn et al., 2017) for validating inner Bayesian learning performance. The overall algorithm is described in Algorithm 5.

---

**Algorithm 5** Simple Bayesian Meta-Learning for Reinforcement Learning

---

1: Initialize $\Theta_0$
2: **for** $t = 0, \ldots$ until converge **do**
3:     Sample a mini-batch of tasks $\mathcal{T}_t$ from $p(\mathcal{T})$
4:     **for** each task $\tau \in \mathcal{T}_t$ **do**
5:         Sample trajectories $\mathcal{D}_\tau^{\text{trn}}$ with $\Theta_0$ in $\tau$
6:         Compute chaser $\Theta_\tau^n = \text{SVPG}(\Theta_0; \mathcal{D}_\tau^{\text{trn}})$
7:         Sample trajectories $\mathcal{D}_\tau^{\text{val}}$ with $\Theta_\tau^n$ in $\tau$
8:     **end for**
9:     $\Theta_0 \leftarrow \Theta_0 - \beta \nabla_{\Theta_0} \sum_{\tau \in \mathcal{T}_t} \mathcal{L}_\tau^{meta}(\Theta_\tau^n; \mathcal{D}_\tau^{\text{val}})$
10: **end for**

---

### C.2.2   SVPG-Chaser

This method is to use SVPG as inner-update and chaser loss for meta-update to maintain uncertainty. Different to supervised learning, this method updates leader particles just with $\mathcal{D}_\tau^{\text{val}}$ in policy gradient update manner. Same chaser loss to supervised learning ones is consistently applied to evaluating the chaser loss extensibility. The chaser loss in RL changes the meta update from a policy gradient problem to a problem similar to imitation learning. Unlike conventional imitation learning with given expert agent, this method keeps the uncertainty provided by the SVPG by following one more updated agent, and ultimately ensures that the chaser agent is close to the expert. Compared to Algorithm 5, this method adds updating the leader and changes the method of meta update like Algorithm 6.

### C.2.3   VPG-Reptile

Reptile (Nichol et al., 2018) solved meta learning problem by using only $1^{\text{st}}$-order derivatives, and used meta update in parameter space. We design a version of meta loss similar to Reptile to verify

---

**Algorithm 6** Bayesian Meta-Learning for Reinforcement Learning with Chaser Loss

---

1: Initialize $\Theta_0$
2: **for** $t = 0, \dots$ until converge **do**
3:     Sample a mini-batch of tasks $\mathcal{T}_t$ from $p(\mathcal{T})$
4:     **for** each task $\tau \in \mathcal{T}_t$ **do**
5:         Sample trajectories $\mathcal{D}_\tau^{\text{trn}}$ with $\Theta_0$ in $\tau$
6:         Compute chaser $\Theta_\tau^n = \text{SVPG}(\Theta_0; \mathcal{D}_\tau^{\text{trn}})$
7:         Sample trajectories $\mathcal{D}_\tau^{\text{val}}$ with $\Theta_\tau^n$ in $\tau$
8:         Compute leader $\Theta_\tau^{n+s} = \text{SVPG}(\Theta_\tau^n; \mathcal{D}_\tau^{\text{val}})$
9:     **end for**
10:    $\Theta_0 \leftarrow \Theta_0 - \beta \nabla_{\Theta_0} \sum_{\tau \in \mathcal{T}_t} D(\Theta_\tau^n || \text{stopgrad}(\Theta_\tau^{n+s}))$
11: **end for**

---

the performance of chaser loss in RL problem. This method computes the chaser $\Theta_\tau^n$ using $\mathcal{D}_\tau^{\text{trn}}$ and then calculates the euclidean distance between this parameter and the global parameter as a meta loss (to prevent the gradient from being calculated through the chaser parameter to maintain the $1^{\text{st}}$-order derivatives). The overall algorithm is described in Algorithm 7.

---

**Algorithm 7** VPG-Reptile

---

1: Initialize $\Theta_0$
2: **for** $t = 0, \dots$ until converge **do**
3:     Sample a mini-batch of tasks $\mathcal{T}_t$ from $p(\mathcal{T})$
4:     **for** each task $\tau \in \mathcal{T}_t$ **do**
5:         Sample trajectories $\mathcal{D}_\tau^{\text{trn}}$ with $\Theta_0$ in $\tau$
6:         Compute chaser $\Theta_\tau^n = \text{VPG}(\Theta_0; \mathcal{D}_\tau^{\text{trn}})$
7:     **end for**
8:     $\Theta_0 \leftarrow \Theta_0 - \beta \nabla_{\Theta_0} \sum_{\tau \in \mathcal{T}_t} D(\Theta_0 || \text{stopgrad}(\Theta_\tau^n))$
9: **end for**

---

## C.3 Experimental Details

Inner update learning rate and the number of inner update are set as 0.1 and 1 for all experiments, which are locomotion (ant/cheetah goal velocity and ant/cheetah goal direction) and 2D-Navigation experiments. Meta update learning rate is set as 0.1 for ant goal direction and 0.01 for other experiments. $\eta$, the parameter that controls the strength of exploration in SVPG is set as 0.1 for ant velocity experiment with SVPG-Chaser, ant goal direction and 2D Navigation with SVPG-Chaser, and 1.0 for other experiments. Each plots are based on an mean and a standard deviation from three different random seed. The subsumed results are plotted with the average reward of maximum task rewards in during of particles.

## C.4 Additional Experiment Results

### C.4.1 2D Navigation

We also compare the models on the toy experiment designed in previous work (Finn et al., 2017), 2D Navigation. This experiment is a set of tasks where agent must move to different goal positions in 2D, which is randomly set for each task within a unit square. The observation is the current 2D position, and actions correspond to velocity clipped to be in the range [-0.1, 0.1]. The reward is the negative squared distance between the goal and the current position, and episodes terminate when the agent is within 0.01 of the goal or at the timestep = 100. We used 10 episodes per each inner update ($K = 10$) and 20 tasks per each meta update. The models are trained for up to 100 meta iterations. The policy network has two hidden layers each with 100 ReLU units. We tested the number of particles for $M \in \{1, 5, 10\}$ with $M = 1$ only for non-ensembled MAML. Same to above locomotion experiments, we compare the models as SVPG-TRPO vs VPG-TRPO and SVPG-Chaser vs VPG-Reptile. As shown in Fig. 5, BMAML showed better performance than EMAML on both comparisons.

**(a)**

**(b)**

**Figure 5:** 2D Navigation: (a) and (b) are results of SVPG-TRPO vs VPG-TRPO and SVPG-Chaser vs VPG-Reptile with three different random seed, respectively.

**(a)** MAML (M=1)      **(b)** EMAML (M=20)      **(c)** BMAML (M=20)

**Figure 6:** Regression qualitative examples: randomly sampled tasks with 10 examples (10-shot) and 10 gradient updates for adaptation