[Reviews · NeurIPS 2018]

Reviewer 1



The paper provides a way to obtain an approximate Bayesian posterior for MAML parameters on a task. The posterior should be proportional to the prior * likelihood. Stein variational gradient descent is used to train a set of particles to approximate the (unnormalized) posterior. The obtained posterior can be used to answer probabilistic queries. The Bayesian method also prevents overfitting when seeing the same training data multiple times. I'm glad I had the honor to read the paper. The method is principled. The explanation is well written. The experiments cover many domains and are well designed. Comments: 1) A minor detail for the prediction construction on line 130: For a general loss, the prediction should be chosen to minimize the expected loss, where the expectation is taken with respect to the posterior distribution. E.g., the absolute error would be minimized by the median instead of the mean. 2) In the experiments, you compare SVPG-Chaser to VPG-Reptile. Why do you say that the chaser loss performs only 1st-order gradient optimization there? Do you use a variant with d(\Theta_0 || \Theta^{n+s}) instead of d(\Theta^n || \Theta^{n+s})? Minor typos: Line 286: VGP should be VPG. Update: Thank you for the answers in the rebuttal. About the comment 1): The Bayesian ensemble of the prediction is not always the best prediction. The best prediction depends on the used loss function (e.g., a loss can be higher for false positives). You should choose the prediction that minimizes the expected loss.

Reviewer 2



Summary: Meta-learning is motivated by the promise of being able to transfer knowledge from previous learning experiences to new task settings, such that a new task can be learned more effectively from few observations. Yet, updating highly parametric models with little amounts of data can easily lead to overfitting. A promising avenue towards overcoming this challenge is a Bayesian treatment of meta-learning. This work, builds on top of recent work that provides a Bayesian interpretation of MAML (model-agnostic-meta-learning). strengths: - This work addresses an important topic and significant contributions within this field would be of interest to many researchers within the NIPS community - The work is well motivated - A novel Bayesian variant of MAML is proposed, with a two-fold technical contribution: - approximating the “task-train” posterior via SVGD (task-train posterior is the posterior over the task optimized parameters). This contribution is a direct extension of (Grant et al 2018) - where the task-train posterior was approximated via a Gaussian distribution. Applying SVGD instead allows for a more flexible and (potentially) more accurate approximation of a highly complex posterior. - Measuring the meta-loss as a distance between two posterior distributions: 1) the task-train posterior obtained from SVGD on training data only and 2) the task-train-posterior obtained from SVGD on training+validation data. This loss has been termed the “chaser loss”. The objective is to optimize meta-parameters such that distance between these two posteriors is minimized. - An extensive evaluation and analysis of the proposed algorithm is performed, covering a variety of learning problems Shortcomings/room-for-improvement: Clarity of presentation of the technical details and some experimental details. - Section 3: the proposed method. I found this section hard to pass for several reasons:
- in the background section you give a very short intro to MAML, but not to Bayesian MAML (as proposed in Grant et al). This creates two major issues: a) when coming to Section 3, the reader has not been primed for the Bayesian version of MAML yet b) the reader cannot easily distinguish what is novel and what is re-deriving previous work. 
 - Section 3: presenting the technical details. You spent very little amount on explaining a few of the steps of deriving your new algorithm. For instance: - Line 107: “we see that the probabilistic MAML model in Eq (2) is a special case of Eq (3) “ -> that is not easily visible at all. Try to give a one sentence explanation - or make it very clear that this connection is shown in Grant et al - To arrive at equation 4) two approximation steps happen (from what I understand):
1) you approximate the task-train posterior via SVGD - this effectively gives you a set of particles theta_0 and theta_tau. (This is actually very unclear - do you maintain particles for both the meta-parameters theta_0 and the task-specific parameters theta_tau?)
2) you now plug in the task-train posterior into equation 2 - but can’t evaluate the integral analytically. You utilize the fact that your posterior is a collection of particles and approximate the integral via a sum.

You do not very clearly derive this and the algorithm 2 has very little detail. It would be way easier to understand your derivation if you could break down this derivation and include a few more details into your algorithm. Note that this part is one your technical contributions, so this is where space should be allocated.
 - Section 3.2 : chaser-loss vs meta-loss. You present a new loss function for your meta-updates and intuitively motivate it very well. However, here your work takes a big step away from the “original MAML” framework. In MAML the meta-loss is , in a sense, a predictive loss on a dataset. Here you move to a loss that measures the distance in posterior-over-task-parameter space. So you change 2 things at once: 1) measuring loss between distributions instead of point estimates 2) measuring loss in “posterior space” vs “predictive space”. This step should be discussed in a bit more detail - why not measure the distance between predictive distributions? What is the rationale here? - Algorithm 3: what’s the stopgrad? - Experiments: Figure 1: What is the MSE over? Is it the mean-squared error that the meta-parameters achieve on D_val? Or is it the MSE after the meta-parameters were updated to a new task with K data points - measured on that new task? - Experiments: Classification: you mention you follow Finn et all 2017 to generate the tasks for this experiment. However, I cannot find the description of what a task is, and how you generate 10K to 800K different tasks on the miniImageNet data set. I also don’t know what the number of training examples per task is? Please improve that explanation. - Experiments: Classification: you mention that you move from 800K to 10K tasks and call that “such a small number of tasks” - that is a very interesting notion of small. When reading this - the first thought I have is that we have replaced the huge number of training observations with a huge number of training tasks. How many examples do you use per task? 10? So a 800K tasks would produce 8M data points?

Reviewer 3



The paper introduces a Bayesian variant of the model-agnostic meta-learning (MAML) algorithm. MAML works by trying to find an "initial" parameter point from which one can adapt well with a few gradient update steps to different tasks. The first extension (Bayesian fast adaptation) generalizes this into an ensemble, where a set of particles is updated using Stein variational gradient descent to approximate task training posteriors and the validation loss that is used for the meta-update. BMAML extends this further such that the "validation" loss is changed to a loss between the (approximate) posteriors of the task-training and task-training-plus-validation distributions. MAML and BMAML (together with ensemble version of MAML) are compared in regression, classification, and reinforcement learning experiments. Paper is mainly excellently written and the method is very interesting extension of MAML that seems to clearly improve the performance in the presented experiments. Yet, I have some confusions about the interpretation of the method. Main comments: (1) The proposed method seems very much tied to a computational algorithm (SVGD) rather than be a model or procedure description in a Bayesian sense (where one doesn't usually need to specify specific computational algorithms, but models and/or utilities). Also, the used number of SVGD steps seems so small that it seems questionable if the distributions have converged such that they could be thought of as even approximately the posteriors. And if they would be ran to convergence, the initial particles would not have an effect anymore? This makes the meaning of, for example, the train-task posterior and it's conditioning on Theta_0 very vague. (Unless I misunderstood some parts of the method.) (2) The interpretation of the initial particles Theta_0 seems somewhat vague in the manuscript (see also comment 1). Can they be interpreted as representing an implicit prior (or is the implicit prior a combination of these and the SVGD instance, as Grant et al.'s interpretation seems to be based on a quick glance, but which point of view is not given in this paper)? (3) The BMAML meta-objective is trying to find a set of initial particles, such that the approx. posterior with task-train data is similar to the approx. posterior of the task-train-plus-validation data. Again from a Bayesian perspective, this sounds a bit weird that one tries to match posteriors where the only difference is that one has more data than the other (and the somewhat vague conditioning on Theta_0). On the other hand, a practical interpretation of the this is that one is trying to fit the initial particles such that adding the validation data to the training data doesn't change the parameter distribution much. (4) Wouldn't another way of trying to avoid the overfitting of Bayesian fast adaption be to use a proper scoring rule or some such loss function that would evaluate the full posterior predictive distribution and require honest uncertainties rather than a point prediction in Equation 5? (5) Why are there no comparisons to Grant et al. paper? Would it be too computationally costly for the experiments here? Also, why isn't the Bayesian fast adaptation method included in the results? It is mentioned that initial experiments found that it might overfit, but it would still be interesting to compare against to understand where the main improvement of BMAML comes from. (6) One mentioned motivation is that Gaussian approx. as in Grant et al. would not be appropriate in many cases (e.g., BNNs). Yet, the number of particles used is rather small and one would not expect it to be able to represent any complicated posteriors very well. (7) Will code be made available? In summary, it seems that my main confusions arise from the interpretation of Theta_0 and the interpretation of the distributions, e.g., p(theta_tau | D^trn_tau, Theta_0), as posteriors while actually the conditioning on Theta_0 seems to require that the inference procedure is not run to convergence, so that they would actually represent the posteriors. Hopefully, the authors could clarify the interpretation of the method (or point out my misunderstandings). Apart from this, I think the method seems to be an interesting extension of MAML and would be a relevant development of this line of work for NIPS. Update after author feedback: I appreciate the authors' careful response and hope that they revise the manuscript in line with it. I'm still a bit conflicted about the approach from a Bayesian point of view, as it explicitly relies on not running the inference procedure (SVGD) for too long (that is, to actual convergence), since then the effect of the initial set of parameters Theta_0, and hence the meta-learning, would be lost (at least in theory, although possibly not in practice fully). And I'm wondering whether the approach could be formulated in some more principled manner with regard to this issue (e.g., the more direct way of using a proper scoring rule; I'm not sure if I agree that this would be a less "principled Bayesian" way of formulating the problem). On the other hand, from a practical point of view, I think the approach is very interesting and, overall, I think the work is certainly worth publishing. And I guess actually the original MAML has much of this same flavour of as it relies on a limited number of gradient updates (but, of course, it's not and not advertised as a Bayesian method).